# Design and Validation of a New Morphing Camber System by Testing in the Price—Païdoussis Subsonic Wind Tunnel

**David Communier, Ruxandra Mihaela Botez * and Tony Wong** 

Laboratory of Applied Research in Active Controls, Avionics and AeroServoElasticity LARCASE, ÉTS, Montréal, QC H3C 1K3, Canada; david.communier.1@ens.etsmtl.ca (D.C.); tony.wong@etsmtl.ca (T.W.)

\* Correspondence: ruxandra.botez@etsmtl.ca

**Abstract:** This paper presents the design and wind tunnel testing of a morphing camber system and an estimation of performances on an unmanned aerial vehicle. The morphing camber system is a combination of two subsystems: the morphing trailing edge and the morphing leading edge. Results of the present study show that the aerodynamics effects of the two subsystems are combined, without interfering with each other on the wing. The morphing camber system acts only on the lift coefficient at a 0° angle of attack when morphing the trailing edge, and only on the stall angle when morphing the leading edge. The behavior of the aerodynamics performances from the MTE and the MLE should allow individual control of the morphing camber trailing and leading edges. The estimation of the performances of the morphing camber on an unmanned aerial vehicle indicates that the morphing of the camber allows a drag reduction. This result is due to the smaller angle of attack needed for an unmanned aerial vehicle equipped with the morphing camber system than an unmanned aerial vehicle equipped with classical aileron. In the case study, the morphing camber system was found to allow a reduction of the drag when the lift coefficient was higher than 0.48.

**Keywords:** morphing camber; morphing trailing edge; morphing leading edge; wind tunnel testing; morphing wing

## 1. Introduction

In an effort to reduce aircraft fuel consumption, researchers and engineers strive to optimize all aircraft flight phases. To this end, many studies have focused on optimizing aircraft weight (structure and materials), engine efficiency, aircraft trajectories, and aerodynamics, etc. The optimization of the aerodynamics of an aircraft is performed by modifying its shapes and/or its surfaces (wings, empennage, etc.). In terms of the shapes of the aircraft surfaces, they could be optimized according to the flight phases [1]. Some research has targeted the modification of the shapes of the surfaces during flight to adapt them to changing conditions and phases.

This article presents a prototype of a Morphing Camber System (MCS) that enables an aircraft wing to modify its shape during flight. The MCS is intended to change the wing's lift, which it does by having the lowest possible impact on the aircraft drag. As result, the morphing of the wing generates a smaller increase in drag than the present increase when classic control surfaces are used. The purpose of the MCS is to reduce the drag of the aircraft by replacing the slats, the aileron and the flaps with a morphing of the internal structure. In this paper, a new method to achieve the morphing of the internal structure is presented. This method consists of using slits on the ribs to make them compliant. Numerous theoretical studies have been carried out on the aerodynamic shape optimization of wing surfaces [2–6]. However, few functional mechanisms have been tested [7]; Blondeau [8] tested a

morphing aspect ratio in wind tunnel; Chanzy tested a morphing trailing edge in wind tunnel and in a flight test with a UAV integrating the morphing system [9]; Pecora et al. tested a morphing trailing edge based on a full-scale wing of a civil regional transportation aircraft [10–12]; Radestock tested a morphing leading edge mechanism [13]; and Woods also tested a morphing trailing edge system using a fish bone concept [14]. Numerical optimization studies are generally hindered by their feasibility from a mechanical and thus, practical standpoint. Conversely, studies or "experimental analysis" on morphing mechanisms allow less aerodynamic improvement than is ideal. The main obstacles in the design and testing of a morphing mechanism are generally given by the weight, the power consumption of the morphing mechanism and the elasticity of the wing surface [15,16]. According to Weissharr [17], the fact that morphing systems are expensive, that morphing aircrafts are heavier than conventional aircrafts, and that morphing systems are complex and require exotic material are myths and demonstrated it by reviewing early morphing aircraft history.

In previous studies, our LARCASE team has developed morphing wing mechanisms which, by modifying the upper surface of the wing, allow the flow transition delay between the laminar and turbulent regimes around the wing [18–24]. This delay has the effect of reducing the drag generated by the wing.

The present study aims to modify the camber of the wing with the aim to increase its lift coefficient and lift to drag ratio(L/D). It is known that when the lift coefficient is increased, the aircraft can fly with a lower angle of attack, thereby reducing its drag.

## 2. Problem statement

The MCS problem is simplified by dividing the wing into its three sections: the leading edge, the wing box and the trailing edge. Such a morphing system needs two actuating mechanisms to morph the leading edge (Morphing Leading Edge (MLE)) and the trailing edge (Morphing Trailing Edge (MTE)), while the wing box remains fixed.

In this research, a NACA0012 airfoil is considered, which will be morphed by the MCS mechanism into a NACA4412 airfoil. This morphing is equivalent to modifying the grey airfoil (NACA0012) to the black airfoil (NACA4412), as shown in Figure 1. In this figure, these two airfoil shapes are each divided into three sections: the leading edge, the center box and the trailing edge. The center box is the same constant shape and is thus un-modified for both airfoils while the leading and trailing edges are different for both airfoils. The NACA0012 and NACA4412 airfoils were considered because they have the same thickness, and the only difference between their shapes lies in their cambers. This is the reason why this new system is called a "Morphing Camber System" (MCS).

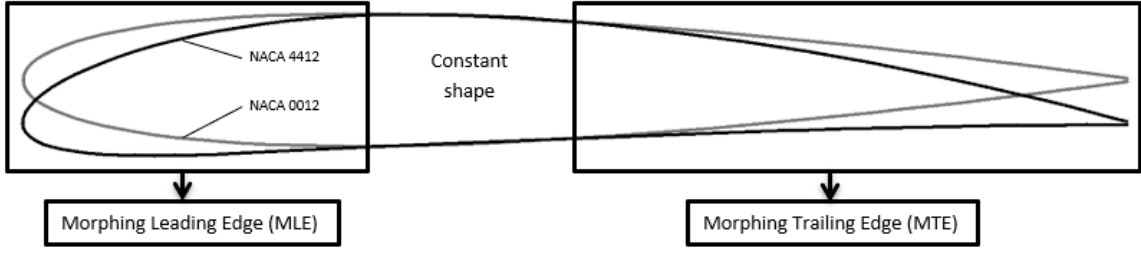

**Figure 1.** NACA0012 and NACA4412 airfoils superimposed.

To compare the efficiency of these two airfoils, an analysis of the L/D ratio is not enough. Indeed, an airfoil could have a lower drag and a higher L/D, while having a lower lift, which could be too low to allow the aircraft to fly. The variation of the drag coefficient with the lift coefficient, allows direct visualization of the airfoil having the smallest drag given a constant lift value. Thus, when comparing the variation of Cl with Cd for NACA0012 airfoil versus the NACA4412 airfoil, it can be seen in Figure 2 that the NACA4412 airfoil is more efficient than the NACA0012 airfoil for a lift coefficient Cl greater than 0.367.

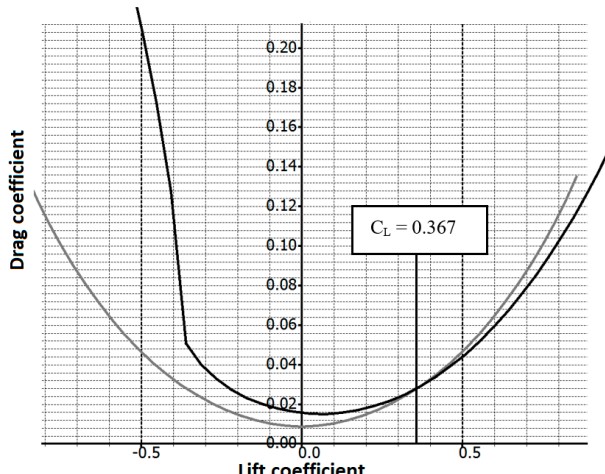

**Figure 2.** Drag coefficient variation with the lift coefficient for NACA0012 (grey) and NACA4412 (black). For Cl = 0.367, angle of attack of NACA0012 = 7.8° and angle of attack of NACA4412 = 2.97°.

This lift coefficient is obtained for an angle of attack of 2.97° for the NACA4412 airfoil, and of 7.8° for the NACA0012 airfoil. It is therefore possible for an aircraft wing, equipped with an MCS, to increase its lift without having to change the angle of attack but only by changing its shape from a NACA0012 to a NACA4412. Furthermore, as the angle of attack of the aircraft could be smaller with an MCS than with a classical wing, there is an additional reduction in drag from other aircraft surfaces (fuselage, empennage).

A measurement of the total efficiency gain when considering all components (wing, fuselage, empennage, etc.) of the UAV is obtained by performing comparative flight tests for a UAV with a conventional wing and the same UAV with a morphing wing.

## 3. Design of the Morphing Camber System (MCS)

The wing will be equipped with flexible ribs to morph its camber. The flexibility of the ribs is made possible using vertical slits. Each slit allows the ribs to perform a small rotation as if a pivot connection had been added to the system. By distributing the slits along the ribs, the wing's overall behavior becomes like the one of a morphing camber system (MCS). The functionality and efficiency of this approach have been demonstrated using the morphing leading edge (MLE) and the morphing trailing edge (MTE) concept developed in [25,26]. The maximum displacement of the MLE and the MTE was determined as follows:

$$\text{Maximum displacement} = \sum_{i=1}^{n} y_i, \tag{1}$$

where:

$$y = \frac{l \times L}{p}, \tag{2}$$

where $n$ is the number of slits, $l$ the width of the slit, $p$ represents the depth of the slit, and finally, $L$ represents the distance between the slit and the end of the rib [25].

Using the slit dimensions given in Tables 1 and 2, a maximum MTE displacement y of 1.33 in (33.78 mm), and a maximum MLE displacement y of 0.322 in (8.18 mm) were obtained using Equations (1) and (2). This displacement corresponds to the one observed on the model before the surface is installed on the wing.

**Table 1.** Slits' dimensions on the trailing edge of the rib.

| Slit Number | $l_n$ (in) | $L_n$ (in) | $p_n$ (in) |
|:---:|:---:|:---:|:---:|
| 1 | 0.026 | 4.911 | 0.416 |
| 2 | 0.026 | 4.635 | 0.4 |
| 3 | 0.021 | 3.609 | 0.34 |
| 4 | 0.021 | 3.338 | 0.317 |
| 5 | 0.014 | 2.317 | 0.233 |
| 6 | 0.014 | 2.053 | 0.205 |

**Table 2.** Slits' dimensions on the leading edge of the rib.

| Slit Number | $l_n$ (in) | $L_n$ (in) | $p_n$ (in) |
|:---:|:---:|:---:|:---:|
| 1 | 0.012 | 0.436 | 0.262 |
| 2 | 0.014 | 0.623 | 0.306 |
| 3 | 0.018 | 0.817 | 0.340 |
| 4 | 0.021 | 1.037 | 0.317 |
| 5 | 0.024 | 1.258 | 0.394 |
| 6 | 0.026 | 1.507 | 0.413 |

The MLE displacement can also be obtained using Finite Element Analysis (FEA) [27] under CATIA V5 with a 3.3% error with respect to its experimental determined value (Figure 3).

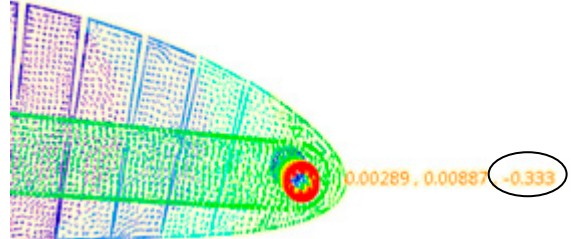

**Figure 3.** MLE tip maximum displacement of 0.333 in (8.46 mm).

The maximum displacement of the MTE was not obtained directly from the FEA, as it was limited by the CATIA V5 software at a maximum rotation of 10° for the MTE displacement (Figure 4).

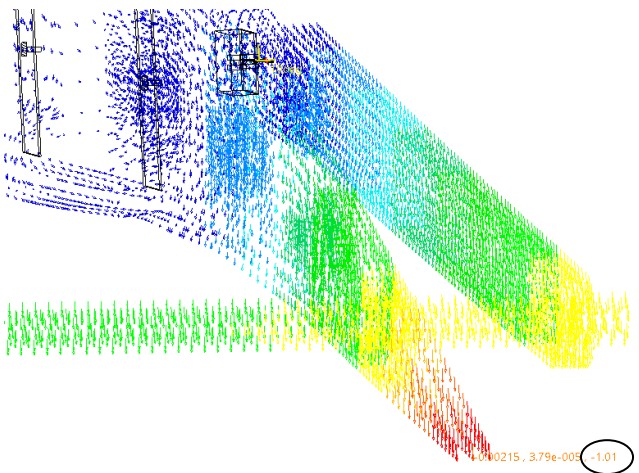

**Figure 4.** MTE tip displacement of 1.01 in (25.65 mm) for a 10° rotation of servomotor.

We note, however, that the displacement of the MTE is linearly proportional to the rotation angle of the servomotor (Table 3). As shown in Table 3, a rotation of 13.4° (angle corresponding to the

calculate maximum displacement of 1.33 in (33.78 mm)) for the MLE tips is obtained for its deformation of 1.357 in (34.47 mm), which corresponds to a 2.06% relative error. This displacement is reduced once the surface is installed on the wing because the heat-shrinkable plastic used to hide the slits folds back on itself, which reduces the width of the slits.

**Table 3.** MTE tip displacement values according to the servomotor angle of rotation.

| Servomotor Angle | Tip Displacement |
|:---:|:---:|
| 0° | 0 in |
| 5° | 0.549 in |
| 10° | 1.01 in |
| 13.4° | 1.357 in |

The impact of each morphing system on the overall wing behaviour is obtained thanks to the two previous studies on the MTE ((Figure 5) [25,26]) and on the MLE (Figure 6).

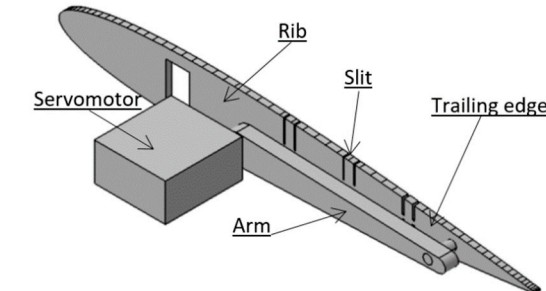

**Figure 5.** MTE system.

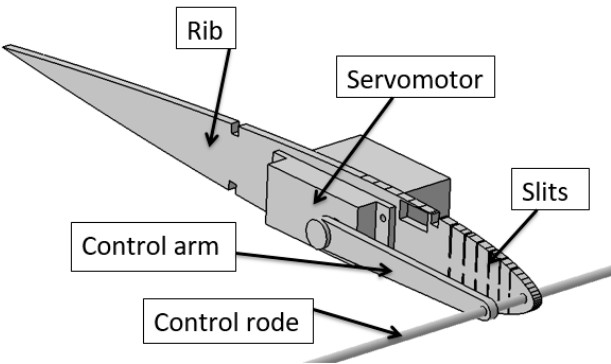

**Figure 6.** MLE system.

The MCS is a combination of these two morphing systems and it maintains the dimensions of the slits for the trailing edge (Table 1) and the leading edge (Table 2), as shown in Figure 7. Consequently, it becomes possible to identify potential interactions between the MTE system and the MLE system aerodynamic performance.

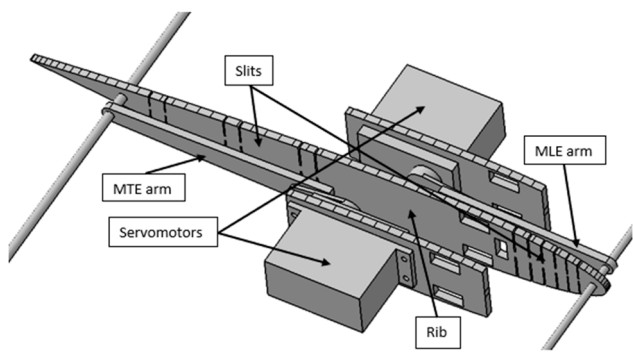

**Figure 7.** MCS design in CATIA V5.

To compare MCS performances with those of a non-morphing system, the prototype of the MCS must have the same dimensions as the non-morphing system (reference wing): 10 in (254 mm) chord, 12 in (304.8 mm) in span. The MCS has a morphing section span of 9.5 in (241.3 mm) with 1 in (25.4 mm) on each side of the morphing section that cannot be morphed. The last 0.5 in (12.7 mm) of remaining span was needed to embed the wing to the circular base of the aerodynamic loading scales. The total span in contact with the airflow was 11.5 in (292.1 mm).

The morphing section of the wing is composed of three ribs, which are connected by rods to its trailing and leading edges. These connections allow the ribs to move in parallel, which results in an identical morphing along the span of the morphing section. To obtain a very good aerodynamic behavior, a heat-shrinkable plastic was installed on the surface of the wing. When this plastic was installed over the slits, the morphing surface moved, maximizing the spacing between the slits and preserving the maximum displacement of morphing for the control surfaces.

Figure 8 shows the internal structure of the wing with its MCS. In previous prototypes (MTE and MLE), the servomotors were fixed on the central rib. The MCS required two servomotors; two servomotor brackets were added next to the central rib on the wing structure. All other features of the design (ribs with slits, main spars, control rods, control arms), as shown in Figure 8, were designed in the MTE, for the rear part of the wing, or in the MLE, for the front part of the wing.

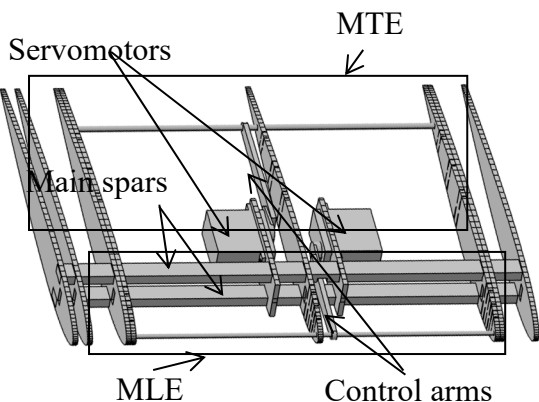

**Figure 8.** MCS internal assembly.

## 4. Prototype Manufacturing

The prototype of the MCS requires wood as raw material, while the other parts are made from commercially available material and hardware. All the wooden pieces, with the sole exception of the spar, were cut with a LASER cutting machine, enabling the pieces to be connected (with glue) together. To ensure the correct positioning of the pieces during gluing, the jigs were also done with plywood and cut using the LASER machine. These jigs provide the exact spacing between the ribs to obtain a

morphing span of 9.5 in (241.3 mm) with an overall span of 12 in (304.8 mm), the angle of incidence of the ribs and the perpendicularity of the ribs according to the main spars (Figure 9).

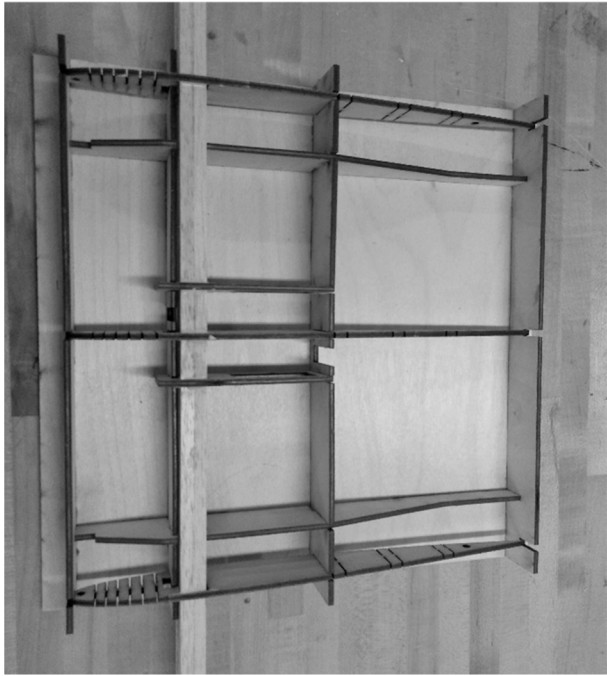

**Figure 9.** Morphing ribs placed in the manufacturing guide.

The prototype was manufactured in two phases. First, the central section was constructed. This section corresponds to the MCS of the wing, with a span of 9.5 in (241.3 mm). Each rectangular part of the wing surface was LASER-cut to ensure the straightest slits possible, as shown in Figure 10.

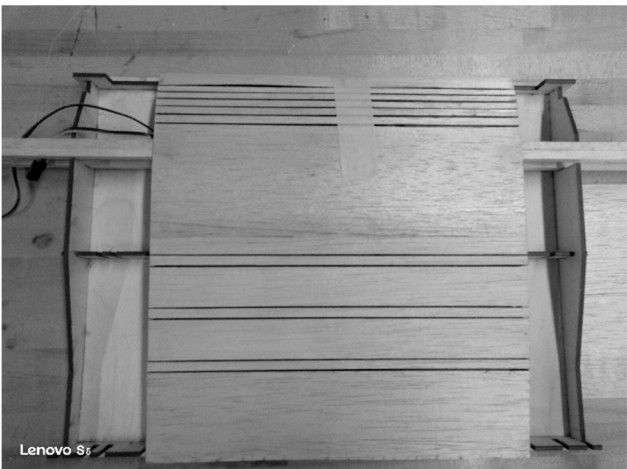

**Figure 10.** Central section of the wing corresponding to the MCS.

In order to obtain the surface at the leading edge, the balsa leaves were moistened and then shaped into molds (Figure 11). The molds were also cut with a LASER cutting machine on plywood.

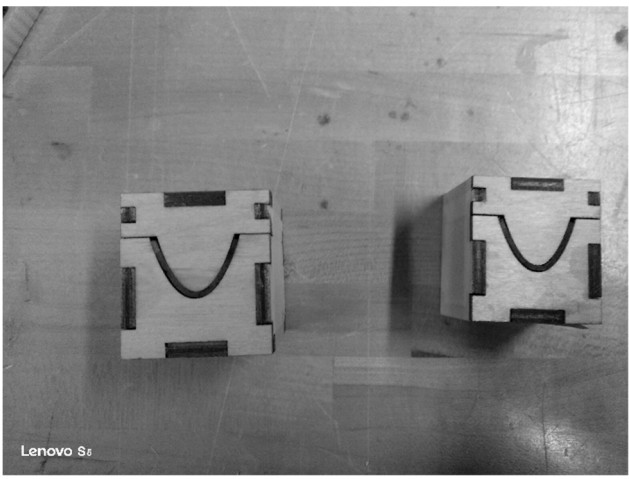

**Figure 11.** Leading edge mold.

## 5. The Price–Païdoussis Wind Tunnel

The LARCASE laboratory is equipped with the Price–Païdoussis subsonic wind tunnel (Figure 12), which has already been used in many LARCASE projects [19,25,28–30]).

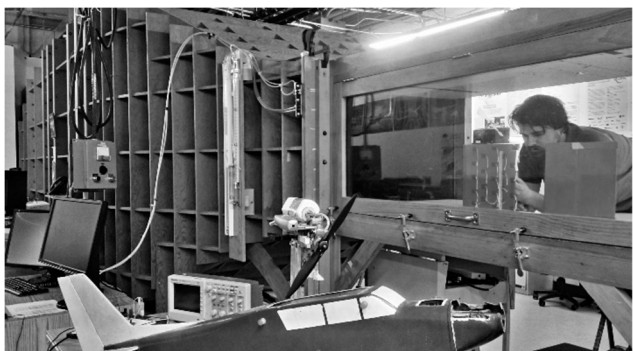

**Figure 12.** Price–Païdoussis subsonic wind tunnel of LARCASE.

This open circuit wind tunnel (Figure 13) is modular and allows test chambers of different sizes. The test chamber used for wing testing with the MCS is 2 ft (0.609 m) tall, 3 ft (0.914 m) wide and 4 ft (1.219 m) long (Figure 14). This test chamber size enables measurement of model aerodynamic performance at speeds ranging from 6 m/s to 35 m/s.

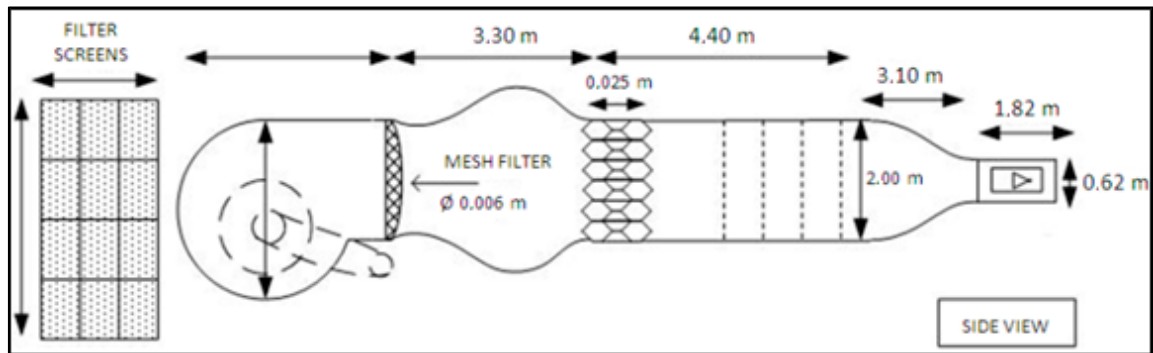

**Figure 13.** Dimensions of the Price–Païdoussis subsonic wind tunnel.

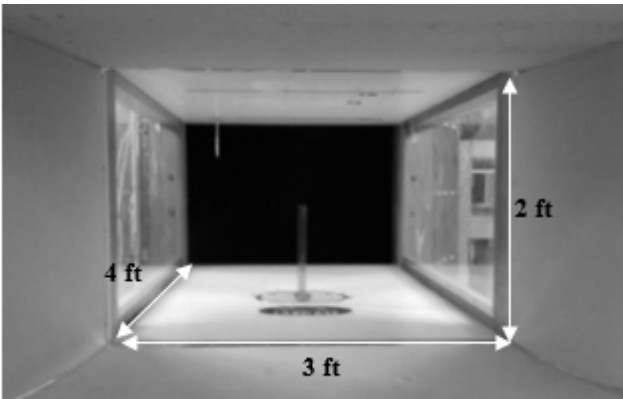

**Figure 14.** Dimension of the test chamber of the wind tunnel.

To obtain results, the wind tunnel was equipped with an aerodynamic loading scale whose design and implementation are explained in [28,31,32] (Figure 15). The aerodynamic loading scales allowed measurement of the forces [31] to control a servomotor installed in the test wing [28] and to control the change of the angle of attack of the wing during testing [32]. For the tests of the wing with the MCS, a servomotor control was added to the interface. The interface had to be modified to allow the display of the forces according to the second servomotor.

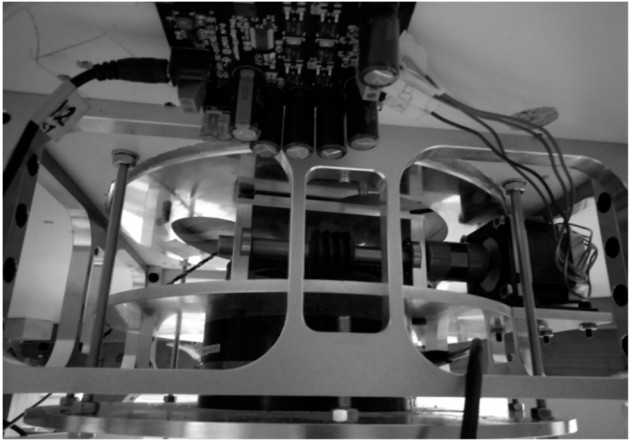

**Figure 15.** Aerodynamic loading scales.

## 6. Results

During wind tunnel tests, the aerodynamic forces of drag and lift were measured on the MCS. All tests were carried out at a speed of 20 m/s, for an air density of 1.18 kg/m$^3$. The leading and trailing edges of the wing were morphed according to their angle values presented in Table 4.

**Table 4.** Morphing cases studied in wind tunnel.

| Case Study | MTE Angle | MLE Angle |
|------------|-----------|-----------|
| Fixed wing | N/A | N/A |
| MCS 0 | 0° | 0° |
| MCS 5 | 5° | 5° |
| MCS 10 | 10° | 10° |
| MTE 5 | 5° | 0° |
| MTE 10 | 10° | 0° |

For each of these cases, the drag (Figure 16) and the lift (Figure 17) forces variation curves with the angle of attack were plotted using measurements carried out in the wind tunnel. From these

measured values, the performance of the wing with the MCS was studied. The drag coefficients' variation showed in Figure 16 that the wing with the MCS generated more drag than a fixed wing (without control surface) for an angle of attack higher than −7° and that its stall angle was lower by 4°. The difference in the variation of the drag coefficient obtained for each morphing case was too small to be analyzed from Figure 16.

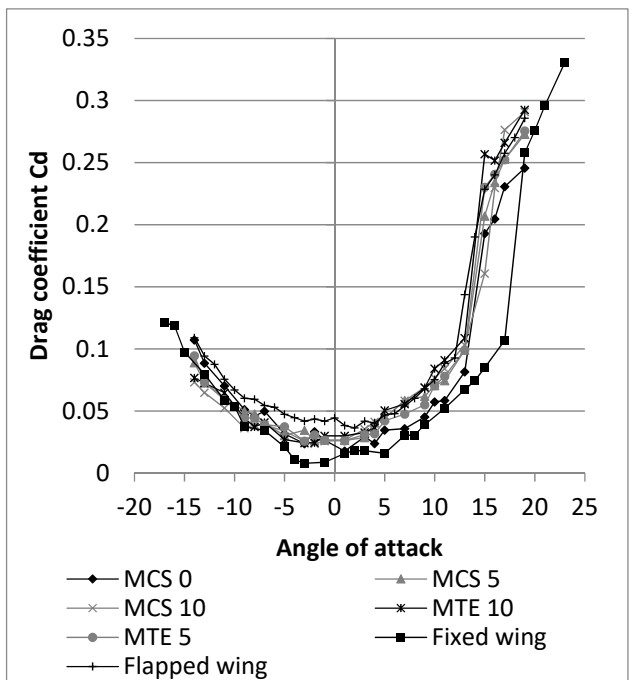

**Figure 16.** Drag coefficient variation with angle of attack.

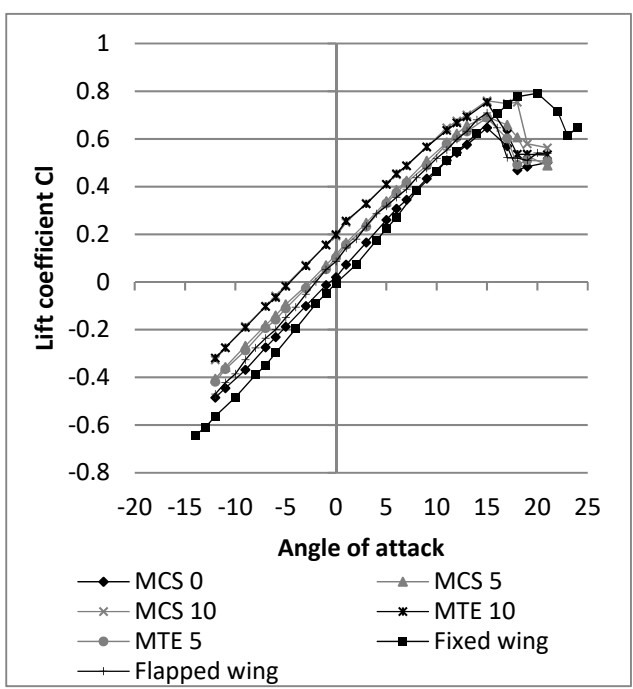

**Figure 17.** Lift coefficient variation with angle of attack.

The lift coefficient variation curves, Figure 17, show that the MTE acts on the variation of the lift coefficient of the wing by causing a shift to the left, for a positive angle. The MLE has no effect on the value of the lift coefficient for an angle of attack of 0° and the slope of the variation of the lift coefficient.

Figure 18 shows that the L/D variations with the angle of attack values are close to each other for each of morphing camber cases. The maximum L/D tends to shift to the left of the curves (closer to the angle of attack 0°) when the morphing of the camber increases. It is also shown that the MCS reduces the L/D of the wing for angles of values of 5° to the stall angle, but for angles smaller than 5°, the MCS gives the highest L/D.

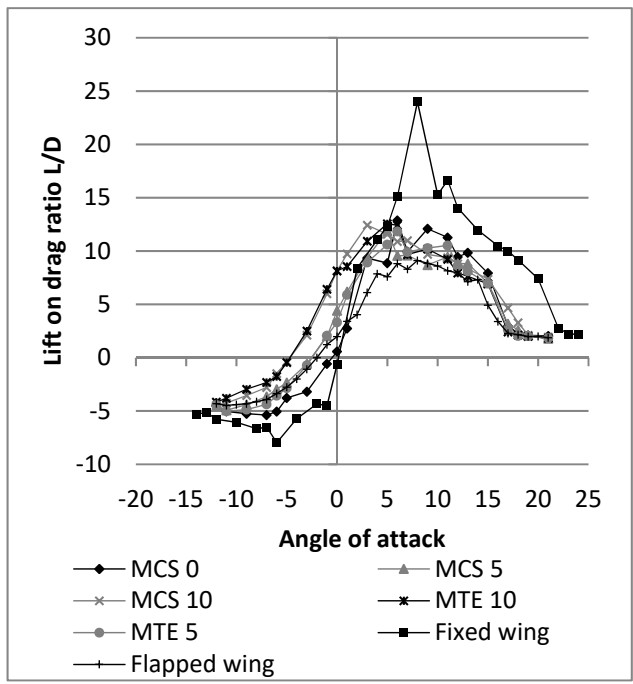

**Figure 18.** Lift on drag ratio variation with the angle of attack.

Figure 19 shows a modification of wing behavior during the stall when camber morphing is at 10° (case MC 10). These behaviors of the drag and lift coefficient variation curves with the morphing of the trailing and leading edges are identical to those observed during independent analyses of the MTE and MLE systems [25]. It can thus be seen that the presence of the MLE system does not generate any interferences on the MTE system and that there is no additional drag due to the combination of the two systems on the same wing.

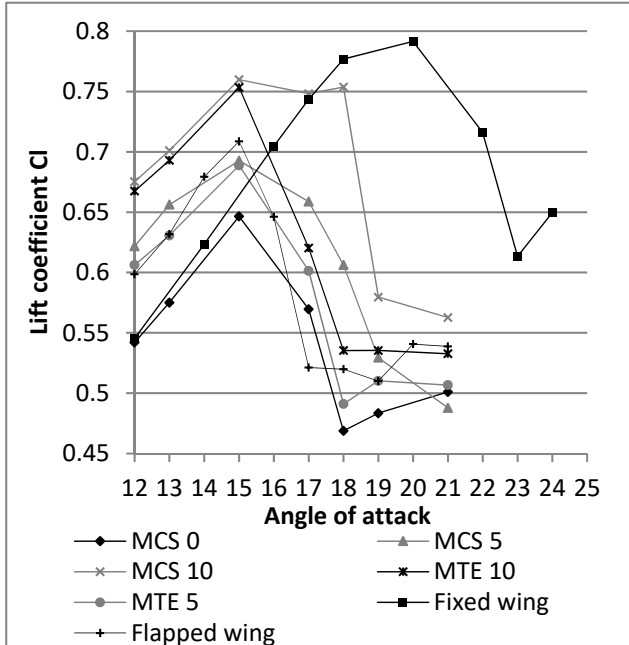

**Figure 19.** Lift coefficient variation with the angle of attack around the stall angle.

The following figures and discussion will only consider the MCS 0, MCS 5, MCS 10, and Fixed Wing cases to improve the visibility of the curves. However, these curves alone cannot assess the effectiveness of the wing with the MCS because of the fact that for a given angle, the MCS provides more lift, but generates more drag, than a fixed wing. In the next section, the methodology to conclude on the effectiveness of the MCS is discussed.

## 7. Discussion

During the design of an aircraft, we seek to obtain a target lift value. To allow an analysis of the performance of the wing with the MCS, it is preferred to represent the drag coefficient variation with the lift coefficient. These curves made it possible to identify the configuration that generates the least drag, given a target lift coefficient value. This idea allows to define its specific morphing configuration for drag minimization, for each flight case of an aircraft, characterized by a needed lift. The drag coefficient variation differences are small between each case study. As we want to establish the effectiveness of the MCS for its use on an aircraft, we can reduce the number of measurements displayed by keeping only the measurement for positive lift coefficients and by removing the measurements corresponding to the stall angles. Thus, in Figure 20, the values obtained for each case studied are compared. It can be observed that a fixed wing allows obtaining the least drag in the range of lift coefficients allowed by the NACA0012 airfoil. However, this wing has no control surface for controlling the aircraft, which means that this configuration is not functional for an aircraft.

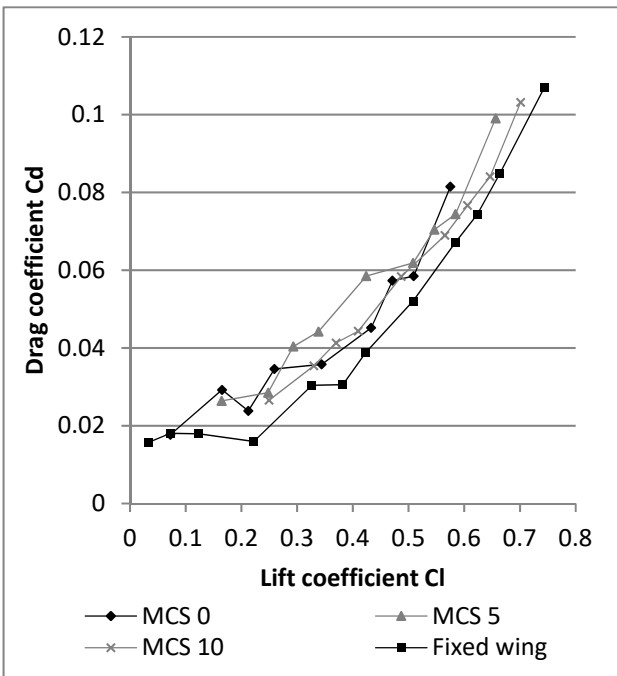

**Figure 20.** Drag coefficient variation with the lift coefficient variation.

The fixed wing configuration was studied for the comparison of its result with those of an MCS 0 configuration. The differences in these two cases for the variation of drag coefficient with the angle of attack (Figure 20) came from the interference between the MCS and aerodynamic fluid around the wing. The variation of the MCS 0 and MCS 10 drag coefficient values are too close for the lift coefficient values between 0.25 and 0.5 to define which configuration would generate less drag than the other configuration. For a lift coefficient value less than 0.1, the MCS 0 case generates less drag; similarly, for a lift coefficient value greater than 0.5, the MCS 10 case generates less drag. There is no lift coefficient value for which the MCS 5 case generates less drag, but for lift coefficient values less than 0.25 and between 0.5 and 0.6, the MCS 5 case is very close to the MCS 10 case. It would therefore be preferred to make the transition between the MCS 0 case and the MCS 10 case when the lift coefficient required to make the aircraft fly will be between 0.5 and 0.6. The drag generated by the wing during a flight will be minimized.

## 8. Application of results with the UAS-S4 Ehécatl

The Unmanned Aerial System (UAS) S4 Ehécatl is an autonomous flight system. It includes an onboard camera, an autopilot and the sensors necessary for its operation. The UAS-S4 Ehécatl model was developed in Mexico by Hydra Technologies in 2002 and made its first flight in 2006. It is therefore a continuous development project for almost 15 years. The UAS-S4 Ehécatl is used by the Mexican army and police. As part of the research developed at the Research Laboratory in active control, avionics and aeroservoelasticity (LARCASE), we use it to develop new methods of analysis, design and manufacturing applied to morphing wings. The geometries for the UAS-S4 in this paper are in Table 5.

**Table 5.** Geometrical data of the UAS-S4.

| Geometry | Value |
|---|---|
| Wing Surface S | 2.9769 m$^2$ |
| Root chord CR | 656.159 mm |
| Wing span b | 4.19 m |
| Wing Taper Ratio λ | 0.6057 |

To measure the impact of the MCS on the drag and lift forces of the UAS-S4, we must calculate the $Cd_0$ and $cl_\alpha$ values from the lift and drag coefficient values using Equation (3).

The value of $cl_\alpha$ can be calculated using the equation:

$$Cl_\propto = \frac{cl_\propto}{1 + \frac{cl_\propto}{\pi * AR * e}},$$

(3)

where the aspect ratio (*AR*) of the wing is:

$$AR = b/\bar{c},$$

(4)

and the Oswald coefficient for a straight wing is:

$$e = 1.78 * \left(1 - 0.045 * AR^{0.68}\right) - 0.64,$$

(5)

The value of $Cl_\alpha$ is obtained from the lift coefficient linear variation curves with the angle of attack (Figure 21), and the equations of these curves are presented in Table 6.

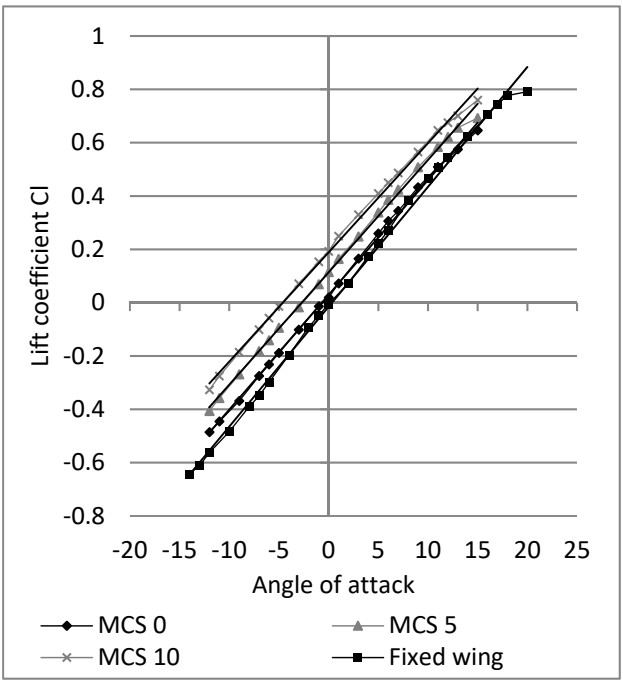

**Figure 21.** Lift coefficient variation curve with the angle of attack.

**Table 6.** Equation of lift coefficient variation with the angle of attack.

| Studied Cases | Cl equation of Variation | $Cl_\alpha$ |
|---|---|---|
| Fixed wing | y = 0.0446x − 0.0112 | 0.0446 |
| MC 0 | y = 0.043x + 0.0294 | 0.043 |
| MC 5 | y = 0.0421x + 0.1138 | 0.0421 |
| MC 10 | y = 0.041x + 0.1885 | 0.041 |

The span of the wing in the wind tunnel was 11.5 in. Because the floor of the wind tunnel acts as a symmetry plane with respect to the aerodynamics flow around the wing, the span considered for the calculation must be 23 in., and the chord has 10 in. The aspect ratio AR = 2.3 and the Oswald coefficient value e = 0.9989 were obtained. These values allow us to calculate the $cl_\propto$ corresponding to the wing airfoil for each case study. The $cl_\propto$ values are presented in Table 7.

**Table 7.** Values of $cl_\alpha$ for each case study.

| Case Study | $cl_\alpha$ |
|---|---|
| Fixed wing | 0.0449 |
| MC 0 | 0.0433 |
| MC 5 | 0.0423 |
| MC 10 | 0.0412 |

The value of $Cd_0$ is calculated using the equation:

$$Cd_0 = Cd - Cd_i, \tag{6}$$

where:

$$Cd_i = \frac{Cl^2}{\pi * AR * e}, \tag{7}$$

From the equations of Cl, the induced drag $Cd_i$ for each case studied was calculated (Figure 22). The value of Cd was obtained using the drag coefficient variation curve with the angle of attack (Figure 23). The equations of these polynomial (2nd order) curves are presented in Table 8. From the induced drag $Cd_i$ and the total drag Cd, the value of drag of the skin friction $Cd_0$ is determined (Figure 24).

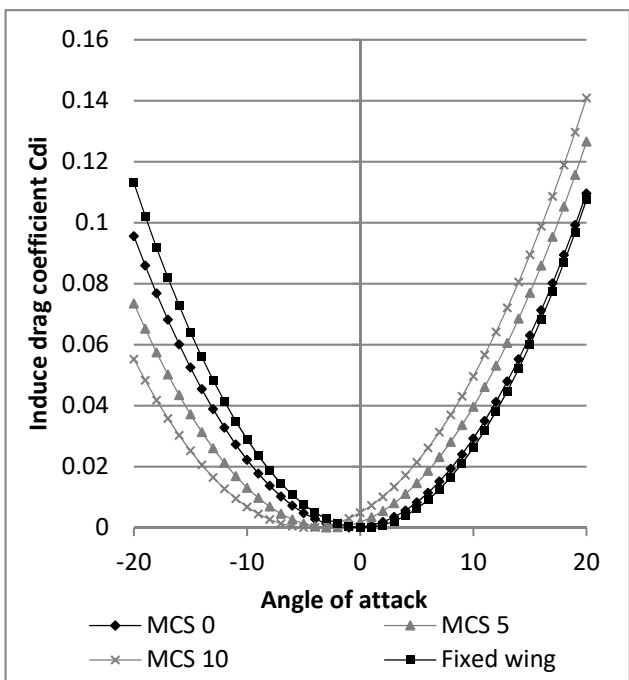

**Figure 22.** Induced drag coefficient $Cd_i$ variation with the angle of attack.

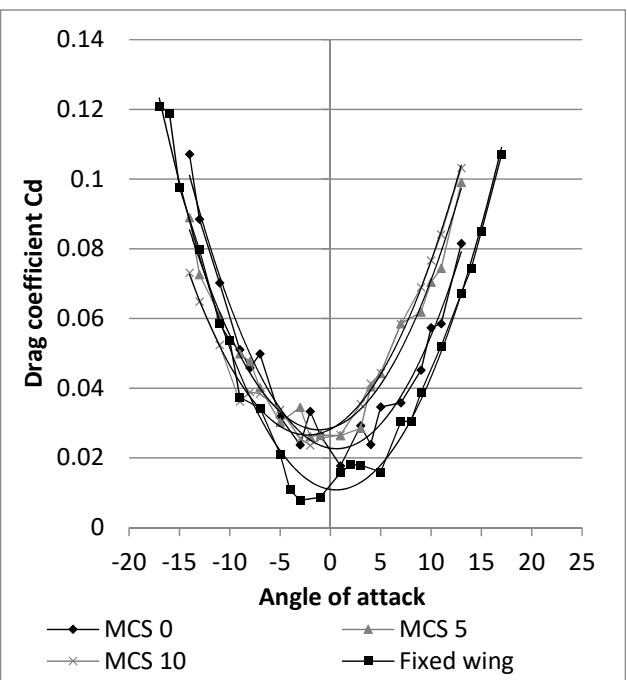

**Figure 23.** Drag coefficient Cd variation with the angle of attack.

**Table 8.** Equation of drag coefficient variation with angle of attack.

| Studied Cases | Cd Equation of Variation |
|---|---|
| Fixed wing | $y = 0.000364x^2 - 0.000397x + 0.010814$ |
| MC 0 | $y = 0.000368x^2 - 0.000446x + 0.022767$ |
| MC 5 | $y = 0.000347x^2 + 0.000788x + 0.028511$ |
| MC 10 | $y = 0.000333x^2 + 0.001496x + 0.028223$ |

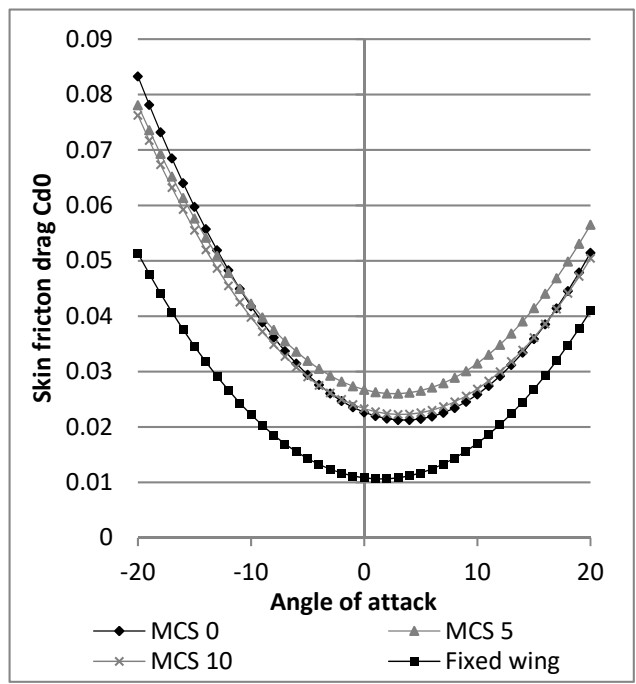

**Figure 24.** Skin friction drag variation with the angle of attack.

As shown in Figure 20, for the same lift value, the fixed wing generates less drag. However, as seen in Figure 17, for the same lift value, the angle of attack of the MCS is lower than that of the fixed wing. This observation implies that the angle of attack of the fuselage for an aircraft using the MCS will be lower than the angle of attack of the fuselage of a conventional aircraft. The variation of the drag coefficient with the lift coefficient is recalculated using the UAS-S4 wing and fuselage geometry as the fuselage geometry needs to be considered. The drag and lift coefficients of the fuselage were obtained using the UAS-S4 in-house simulator for flight conditions corresponding to the wind tunnel tests (20 m/s) [33]. The drag and lift performance of the wings were obtained by using Equations (3), (4) and (8) to (10) with the previously calculated $cl_\alpha$ and $Cd_0$ values, and by use of the wing geometry of UAS-S4.

The equation for modeling the drag coefficient of the fuselage with the angle of attack (AoA) from the drag coefficients used in the simulator at a speed of 20 m/s is:

$$Cd_{0f} = 1.442834 * 10^{-6} * AoA^3 + 7.472932 * 10^{-5} * AoA^2 - 3.14648307 * 10^{-4} * AoA + 4.434891635 * 10^{-3} \tag{8}$$

The equation for modeling the lift coefficient of the fuselage with the angle of attack (AoA) from the drag coefficients used in the simulator at a speed of 20 m/s is:

$$Cl_f = 3.08 * 10^{-6} * AoA^3 + 6.872 * 10^{-5} * AoA^2 + 3.33018 * 10^{-3} * AoA - 8.16467 * 10^{-3} \tag{9}$$

In Equations (8) and (9), the angle of attack (AoA) is in degrees.

The drag and lift coefficients of the fuselage were expressed according to the surface of the wing. This allows adding the coefficients of the fuselage to the coefficients of the wings. Thus, we could estimate the total drag and lift coefficients according to the following equations:

$$Cl_t = Cl + Cl_f, \tag{10}$$

$$Cd_t = Cd_0 + Cd_{0f} + Cd_i + Cd_{if}, \tag{11}$$

where:

$$Cd_{if} = \frac{Cl_f^2}{\pi * AR * e}, \tag{12}$$

Equation (13) will be used for calculating the Oswald coefficient [1] because of the fact that the wing of UAS-S4 is trapezoidal with a sweep angle $\Lambda_{LE} = 6.353°$ and $AR = 10.4785$.

$$e = 4.61 * \left(1 - 0.045 * AR^{0.68}\right) * \left(\cos \Lambda_{LE}\right)^{0.15} - 3.1, \tag{13}$$

By use of Equation (3) for the calculation of $Cl_\alpha$ of the wing and the values of $Cl_0$ of the test wings, the variation curves of the lift coefficient of the wing with the angle of attack can be traced in Figure 25. Then, Equations (9) and (10) give the lift coefficient variations for the wing and fuselage assembly (Figure 26).

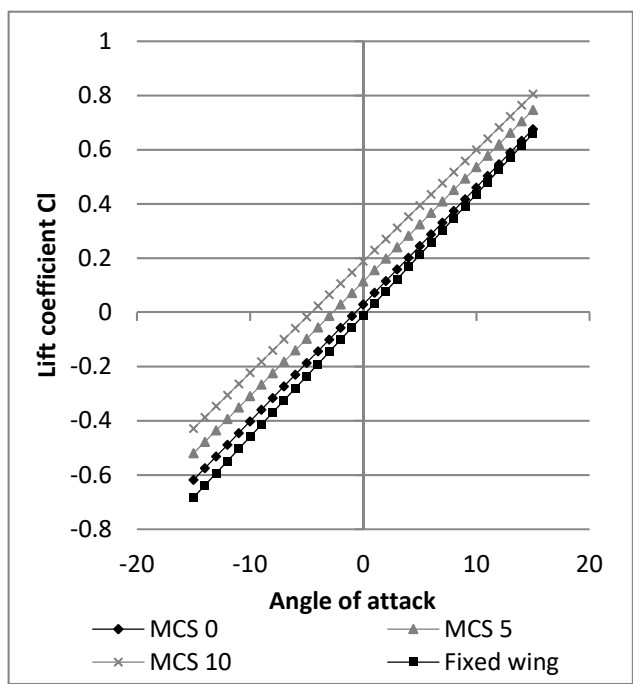

**Figure 25.** Lift coefficient variation with angle of attack for the UAS-S4 wing.

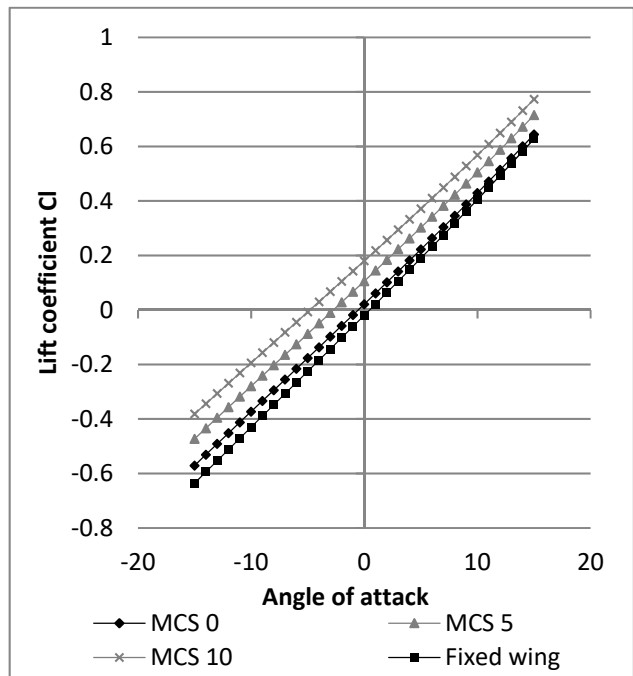

**Figure 26.** Lift coefficient variation with angle of attack for UAS-S4 (wing + fuselage).

Equation (7) allows the calculation of the induced drag coefficients for the UAS-S4 wing from the lift coefficients of the wing. Then, with Equation (11), the combination of wing and fuselage drag coefficients of the UAS-S4 are plotted (Figure 27).

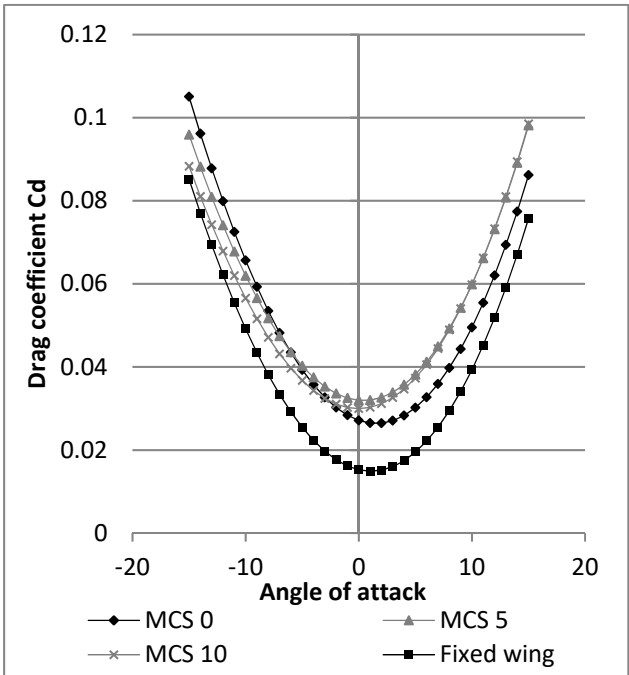

**Figure 27.** Drag coefficient variation with angle of attack for UAS-S4.

Once the drag and lift coefficient variation curves are obtained, the graph representing the variation of the drag coefficient with the lift coefficient is plotted (Figure 28). As seen in Figure 29, for a lift coefficient greater than 0.48, the wing with a morphing of the camber of 10° (MCS 10) generates less drag than a fixed wing. This value corresponds, in Figure 26, to an angle of attack of 11° for a fixed wing and of 7° for a wing with a morphing of the camber of 10°. This observation indicates that for the aircraft lift coefficient greater than 0.48, the MCS will be more advantageous than the fixed wing.

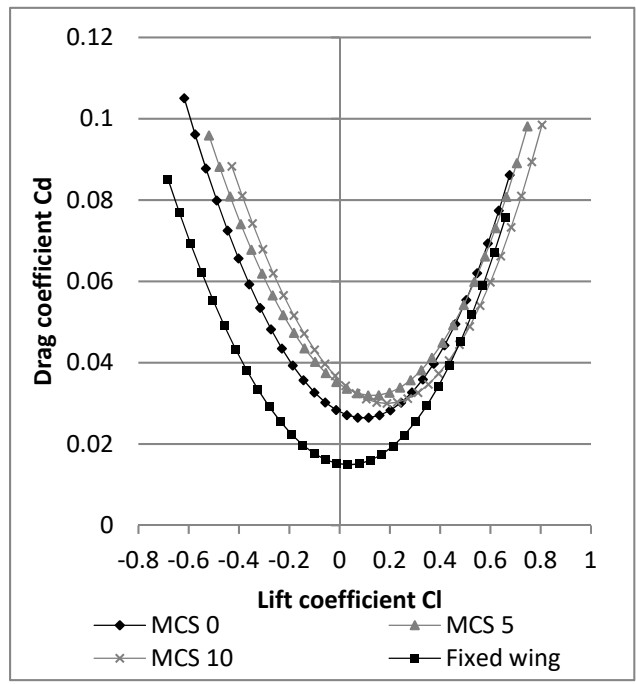

**Figure 28.** Drag coefficient variation with lift coefficient for the UAS-S4.

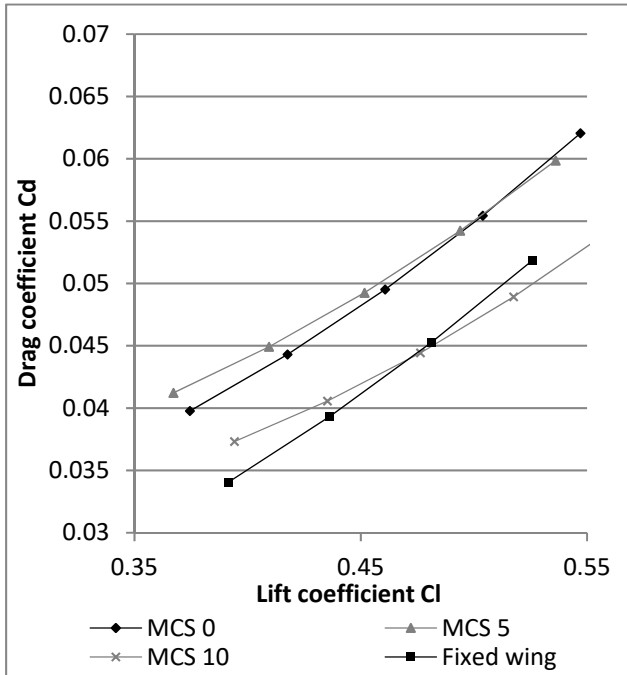

**Figure 29.** Drag coefficient variation with lift coefficient for UAS-S4 (zoom).

These values were calculated for a wing with a NACA0012 airfoil whose performances were obtained in the wind tunnel. This is not the airfoil used in the current UAS-S4, but it is a close one; the values are useful for potential gain that can be provided using an MCS on the UAS-S4.

## 9. Conclusions

In this article, a new system allowing the complete morphing of the camber, from the airfoil leading edge to its trailing edge, was presented from its design to its validation in a wind tunnel. The design of this system provides an easy-to-manufacture and lightweight MCS for the wing structure. The MCS will consequently not penalize the performance of the aircraft from its weight point of view. The method used to calculate the maximum displacements of the MLE and of the MTE makes it easy to size the slits without having to perform an FEA optimization process. In order to apply this method, the thickness of the material remaining at the slits must allow the rib to bend without causing its plastic deformation. To ensure this bending, anisotropic materials, such as wood or carbon fibers in the direction of the length of the rib will be more effective than isotropic materials, such as steel or aluminum. The wind tunnel tests, despite having a lower than expected maximum morphing capabilities due to the manufacturing method, showed that the MCS improved the aerodynamic performance of the wings and of the UAS-S4. The MTE and MLE systems were also shown to be independent. The MTE acted on the lift of the wing, and the MLE acted on the stall angle, thus independent control of both systems was allowed. Wind tunnel performance scaling was used to determine the potential gain from using an MCS on the UAS-S4. The next step will be to design an MCS with the dimensions used for the UAS-S4 wing.

**Author Contributions:** Investigation, methodology, writing, D.C.; Supervision, Writing—review & editing, R.M.B. and T.W. All authors have read and agreed to the published version of the manuscript.

**Funding:** This research received no external funding.

**Acknowledgments:** The authors would like to sincerely thank the Natural Sciences and Engineering Research Council of Canada (NSERC) for the Canada Research Chair Tier 1 in Aircraft Modelling and Simulation New Technologies funding, as well as the McGill University Michael Païdoussis and Stuart Price for the donation of the Price–Païdousis wind tunnel at the LARCASE, ÉTS.

**Conflicts of Interest:** The authors declare that there is no conflict of interest regarding the publication of this paper.

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
