# Peer review of "Design and Validation of a New Morphing Camber System by Testing in the Price—Païdoussis Subsonic Wind Tunnel"

_aerospace, doi:10.3390/aerospace7030023_

Round 1
Reviewer 1 Report
1. This paper reports the wind tunnel testing of a morphing camber design proposed by the authors. The paper is welcome - although there are some experimental papers concerned with morphing, we need to have more so that morphing concepts may be developed into practical demonstrators. I also like the fact the authors report the drag for a given lift, as this is the only sensible way to report results for morphing wings. I have to say that the results only show a relatively marginal benefit for the morphing solution proposed, and it would have been better if results for a conventional flap were also included (see point 2 below). But overall this is an interesting study.
2. My major concern is the comparisons made for the morphing camber solution. The competitor seems to be a NACA0012 with increased angle of attack, and the results show that the NACA0012 generally performs better. But I don't think this is the correct comparison as the NACA0012 does not have a control surface (as indeed stated by the authors). It seems to me that a better comparison is a NACA0012 with a conventional control surface. If the authors had included this comparison then I think the morphing camber solution would be shown to perform much better (and indeed Ref. [14] of the current paper does demonstrate this). Given the time and resources required for wind tunnel testing, I don't think it reasonable for the authors to include a flapped aerofoil, but some discussion should be added.
3. The authors use a heat-shrinkable plastic as a skin for the aerofoil. What is the modulus / stiffness of this skin? Given that the skin above the slits requires a high strain I am not convinced that this is a suitable skin material? Perhaps the authors could comment, and maybe add some discussion to the paper about alternative materials for the skin?
4. The introduction is very short and a literature review section included - I would combine these two sections into the introduction.
5. The authors use Imperial units throughout, whereas SI units are usually preferred, The authors should check the requirements of the journal in terms of units. There are no units given in Tabes 1 and 2.
Author Response
The authors would like to thank to the reviewer 1 for her or his comments that contributed to the improvement of this paper writing. Please find below our answers to his or her comments, the authors
Comment 1.1
This paper reports the wind tunnel testing of a morphing camber design proposed by the authors. The paper is welcome - although there are some experimental papers concerned with morphing, we need to have more so that morphing concepts may be developed into practical demonstrators. I also like the fact the authors report the drag for a given lift, as this is the only sensible way to report results for morphing wings. I have to say that the results only show a relatively marginal benefit for the morphing solution proposed, and it would have been better if results for a conventional flap were also included (see point 2 below). But overall this is an interesting study.
Answer 1.1
The results for a ‘Flapped wing’, which can be considered a ‘conventional flap’ were added in Figures 16-19 L216-L236.
Comment 1.2
My major concern is the comparisons made for the morphing camber solution. The competitor seems to be a NACA0012 with increased angle of attack, and the results show that the NACA0012 generally performs better. But I don't think this is the correct comparison as the NACA0012 does not have a control surface (as indeed stated by the authors). It seems to me that a better comparison is a NACA0012 with a conventional control surface. If the authors had included this comparison then I think the morphing camber solution would be shown to perform much better (and indeed Ref. [14] of the current paper does demonstrate this). Given the time and resources required for wind tunnel testing, I don't think it reasonable for the authors to include a flapped aerofoil, but some discussion should be added.
Answer 1.2
L216-L236: The lift, drag and lift on drag ratio variations with the angle of attack are now presented on Figures 16-19 for the ‘NACA0012 wing with an aileron’ (‘Flapped wing’).
Comment 1.3
The authors use a heat-shrinkable plastic as a skin for the aerofoil. What is the modulus / stiffness of this skin? Given that the skin above the slits requires a high strain I am not convinced that this is a suitable skin material? Perhaps the authors could comment, and maybe add some discussion to the paper about alternative materials for the skin?
Answer 1.3
The skin is not tight above the slits, as it should not influence the deformation of the wing. For this reason, we do not need to take into account the skin stiffness.
We are aware that new research aims to develop flexible skin for morphing wing applications, but we needed to develop a skin easily to manufacture, and at low cost in order to test it in our wind tunnel.
Comment 1.4
The introduction is very short and a literature review section included - I would combine these two sections into the introduction.
Answer 1.4
Following your comment, the two sections ‘introduction’
and ‘literature review’ have been combined.
Comment 1.5
The authors use Imperial units throughout, whereas SI units are usually preferred, The authors should check the requirements of the journal in terms of units. There are no units given in Tabes 1 and 2.
Answer 1.5
The SI units were added for each value presented in Imperial units (L109, L120, L122, L125, L126, L143-L147, L168, L173, L188, L189).
Thus, SI units were added for values presented in Tables 1 and 2.
Reviewer 2 Report
The paper is interesting; however, it raises quite a few questions and comments that need to be addressed:
The idea of changing the lift or drag characteristics of a wing without changing the angle of attack is not a new concept. It has been implemented on nearly every aircraft since the early 1900s, i.e. slats and flaps. If the novelty of the paper is the validation of a particular mechanism via test, the authors need to be clearer on that and what the main contribution of the paper is.
The authors mention that using the L/D as an optimization function may not be sufficient since it does not give a sense for the absolute L. To do this, all they have to do is simply plot the L/D vs L. Many plots are presented in this paper, but none are L/D. Also, given that the change in lift or drag is relatively small between these variations, the argument about the absolute value of L not being presented in L/D vs AoA, although valid, is not a very strong one. I highly suggest adding L/D plots to show the differences between the variations.
The authors need to use more standard terminology when describing commonly knows aircraft features. For instance, in line 85, the authors talk about additional drag associated with an 'aileron' while discussing the AoA. The aileron is a control surface used for controlling the rolling moment, and although it will have an impact on changing the lift or drag characteristics of the wing, it is not typically intended for such applications, unlike devices such as flaps that do so by changing the camber, and modifying the effective AoA. Did you mean to say Flap? If so, please clarify and reword. If you are actually referring to a control surface, such as an aileron, then please elaborate on the details of the particular UAV or aircraft. Is the aileron intended for multiple applications?
With that said, if the intention of the authors is to use the camber change as a means to control the rolling moment, then it would be fair to discuss the aileron in this comparison. However, if that is the case, then this paper should be re-written with a focus on controls rather than performance (L/D). In that case the speed at which the system can be deployed becomes critical. Please clarify.
The data presented in this paper suggests that the Fixed Wing (NACA 0012) outperforms the proposed design at all AoAs. What about a fixed wing NACA 4412? Was that also evaluated? Having the fixed version of the airfoil that the morphing system is trying to achieve would provide some baseline for all the comparisons. However, I don’t see that presented in the paper. This makes it difficult to determine how much of the loss in performance could be attributed to the morphing system itself by means of additional drag or flow separation, as opposed to it being attributed to the actual airfoil shape.
Furthermore, in line 240, the authors mention that the fixed wing is not functional because it has no control surface. Does that mean that the other configurations have a control surface? None of the figures or the text mentions this. If so, please elaborate. If not, then is the camber being used for control? Elaborate. Also, please note that a fixed wing can also be used for control by changing the AoA of each wing independently at the root. In summary, the control question notwithstanding, a fixed wing seems to be outperforming all the configurations at all AoAs when it comes to Drag performance and it offer the largest range of AoA (Fig 19). The authors need to elaborate more about how the proposed system outperforms a conventional design.
Author Response
The authors would like to thank to the reviewer 2 for her or his comments that contributed to the improvement of this paper writing. Please find below our answers to his or her comments, the authors
Comment 2.1
The paper is interesting; however, it raises quite a few questions and comments that need to be addressed: The idea of changing the lift or drag characteristics of a wing without changing the angle of attack is not a new concept. It has been implemented on nearly every aircraft since the early 1900s, i.e. slats and flaps. If the novelty of the paper is the validation of a particular mechanism via test, the authors need to be clearer on that and what the main contribution of the paper is.
Answer 2.1
We added the following explanation on L38-L41: The purpose of the MCS is to reduce the drag of the aircraft by replacing the slats, the aileron and the flaps with an internal morphing structure. In this paper, a new method for the design and manufacture of the internal morphing structure is presented. This method consists in using slits on the ribs in order to make them compliant.
Comment 2.2
The authors mention that using the L/D as an optimization function may not be sufficient since it does not give a sense for the absolute L. To do this, all they have to do is simply plot the L/D vs L. Many plots are presented in this paper, but none are L/D. Also, given that the change in lift or drag is relatively small between these variations, the argument about the absolute value of L not being presented in L/D vs AoA, although valid, is not a very strong one. I highly suggest adding L/D plots to show the differences between the variations.
Answer 2.2
L235: A graph representing the L/D variation with the angle of attack has been added in Figure 18.
L223: Following explanation was added:
Figure 18 shows that the L/D variations with the angle of attack values are close to each other for each of morphing camber cases. The maximum L/D tends to shift to the left of the curves (closer to the angle of attack 0°) when the morphing of the camber increases. It is also shown that the MCS reduces the L/D of the wing for angles of values of 5° to the stall angle, but for angles smaller than 5°, the MCS gives the highest L/D.
Comment 2.3
The authors need to use more standard terminology when describing commonly knows aircraft features. For instance, in line 85, the authors talk about additional drag associated with an 'aileron' while discussing the AoA. The aileron is a control surface used for controlling the rolling moment, and although it will have an impact on changing the lift or drag characteristics of the wing, it is not typically intended for such applications, unlike devices such as flaps that do so by changing the camber, and modifying the effective AoA. Did you mean to say Flap? If so, please clarify and reword. If you are actually referring to a control surface, such as an aileron, then please elaborate on the details of the particular UAV or aircraft. Is the aileron intended for multiple applications?
Answer 2.3
L89: The word ‘classical aileron’ was replaced by ‘classical wing’ in order to clarify the meaning of the sentence.
Comment 2.4
With that said, if the intention of the authors is to use the camber change as a means to control the rolling moment, then it would be fair to discuss the aileron in this comparison. However, if that is the case, then this paper should be re-written with a focus on controls rather than performance (L/D). In that case the speed at which the system can be deployed becomes critical. Please clarify.
Answer 2.4
The MCS presented in this article aims to improve the performance of the aircraft. As presented in our previous article mentioned as ref. [26] in this paper, the MCS can control the aircraft by using the wing trailing edge.
Comment 2.5
The data presented in this paper suggests that the Fixed Wing (NACA 0012) outperforms the proposed design at all AoAs. What about a fixed wing NACA 4412? Was that also evaluated? Having the fixed version of the airfoil that the morphing system is trying to achieve would provide some baseline for all the comparisons. However, I don’t see that presented in the paper. This makes it difficult to determine how much of the loss in performance could be attributed to the morphing system itself by means of additional drag or flow separation, as opposed to it being attributed to the actual airfoil shape.
Answer 2.5
We have not tested a wing with a NACA4412 airfoil, as we installed the MCS on a wing with a NACA 0012 airfoil. A test on a wing with NACA4412 airfoil will not show us the lost due to the MCS, as the fixed wing will still have a NACA0012 airfoil. The additional drag can be obtained by comparing the drag on the NACA0012 wing with the drag on the MCS at 0 degree.
Comment 2.6
Furthermore, in line 240, the authors mention that the fixed wing is not functional because it has no control surface. Does that mean that the other configurations have a control surface? None of the figures or the text mentions this. If so, please elaborate. If not, then is the camber being used for control? Elaborate. Also, please note that a fixed wing can also be used for control by changing the AoA of each wing independently at the root. In summary, the control question notwithstanding, a fixed wing seems to be outperforming all the configurations at all AoAs when it comes to Drag performance and it offer the largest range of AoA (Fig 19). The authors need to elaborate more about how the proposed system outperforms a conventional design.
Answer 2.6
The control surface for the other configuration is the MCS itself. One of its goals is to remove the aileron of the wing, and to control the aircraft using the morphing trailing edge, as mentioned in ref [26]. The use of a rotation of the wing at its root would imply a need for a stronger structure at its root, and for a heavier wing. To be able to compare the configurations, the structure and weight of the wing need to be the same for each test wing. As mentioned in the previous answer, the NACA0012 wing is an optimal reference for a NACA0012 shape in order to quantify the impact of the MCS structure on the wing.
Reviewer 3 Report
Dear Mr. Communier, Dr. Botez, and Dr. Wong:
I read your manuscript with great interest as morphing wing systems are fascinating. Your study is intriguing and is of practical use to the field. Unfortunately, the presentation of your study and the results is difficult to follow. I have enumerated my comments below by line number in the original manuscript.
10 - "morphing camber system", system is missing
41 - "morphing trailing edge in a wind tunnel", wing is spelt as wing
66 - "equivalent to modifying"
86 - "additional reduction"
108/109 - The statement that appears at these line numbers is confusing. The FEA results are not presented in this paper, thus the limitation that the FEA imposes on the MTE displacement is unclear. This statement should be elaborated.
102 - 115 - Several cross-references to figures and tables are made; however, these objects appear over the next page. The text should be re-arranged and the objects interspersed to improve the flow of the explanation.
139 - "section that cannot", no space between can and not
163 - See comment for line 102-115
168 - "The molds were cut" Be careful about verb tense because the tense often shifts between past and present. A thorough re-reading is suggested.
180 - See comment for line 102-115
182 - "This test chamber size enables measurement of"
190 - "allowed measurement of the forces"
207 - See comment for line 102-115
208 - "variation curves, Figure 17, show that"
211 - See comment for line 102-115
234 - See comment for line 102-115
258 - This is the first mention of UAS-S4. The application of the wind tunnel results to the UAS is very unclear within the manuscript. Up to this point in the manuscript, the impression is that only wind tunnel tests were conducted for the morphing camber system applied to a single wing. The UAS aspect needs to be incorporated throughout, with a clear distinction between procedures and results from the wind tunnel, and procedures and results from the UAS simulation.
272/273/274 - There's a typo, "cl@" appears
277 - The theory that appears in this section should be moved earlier. Figures 22 to 24 seem to show results, so moving to the results section is more appropriate.
300 - 304 - These equations should appear earlier in the methodology or theory sections.
315 - Figure 25 has a typo for the x-axis label "angle d'attack"
319 - "Equation (7) allows the calculation of"
Author Response
The authors would like to thank to the reviewer 3 for his or her comments that contributed to the improvement of this paper writing. Please find below our answers to the reviewer 3 comments, the authors
Comment 3.1
L10 - "morphing camber system", system is missing
L41 - "morphing trailing edge in a wind tunnel", wing is spelt as wing
L66 - "equivalent to modifying"
L86 - "additional reduction"
L139 - "section that cannot", no space between can and not
L182 - "This test chamber size enables measurement of"
L190 - "allowed measurement of the forces"
L208 - "variation curves, Figure 17, show that"
L272/273/274 - There's a typo, "cl@" appears
L315 - Figure 25 has a typo for the x-axis label "angle d'attack"
L319 - "Equation (7) allows the calculation of"
Answer 3.1
The suggested changes were done in the revised paper, as follows:
L10: “morphing camber system”
L45: “wind tunnel”
L70: “equivalent to modifying”
L90: “additional reduction”
L145: “section that cannot”
L189: “This test chamber size enables measurement of”
L197: “allowed measurement of”
L219: “variation curves, Figure 17, show that”
L288-290: "cl@" changed for “”
L333: x-axis label change for “Angle of attack”
L335: “Equation (7) allows the calculation of”
Comment 3.2
L108/109 - The statement that appears at these line numbers is confusing. The FEA results are not presented in this paper, thus the limitation that the FEA imposes on the MTE displacement is unclear. This statement should be elaborated.
Answer 3.2
L115-L118: In order to clarify the statement, the sentence “The maximum displacement of the MTE is not obtained directly from the FEA because of the fact that the FEA gives results for a maximum rotation of 10° of the MTE displacement” was replaced with “The maximum displacement of the MTE was not obtained directly from the FEA, , as it was limited by the CATIA V5 software at a maximum rotation of 10° for the MTE displacement”
Comment 3.3
L102 - 115 - Several cross-references to figures and tables are made; however, these objects appear over the next page. The text should be re-arranged and the objects interspersed to improve the flow of the explanation.
L163 - See comment for line 102-115
L180 - See comment for line 102-115
L207 - See comment for line 102-115
L211 - See comment for line 102-115
L234 - See comment for line 102-115
Answer 3.3
As suggested, we changed places in the text for: Table 1 (L112), Table 2 (L113), Figure 3 (L119), Figure 4 (L121), Figure 5 (L132), Figure 6 (L134), Figure 9 (L171), Figure 10 (L175), Figure 16 (L215), Figure 17 (L217), Figure 18 (L230), Figure 19 (L236), Figure 20 (L258) and Figure 21 (L260).
Comment 3.4
L168 - "The molds were cut" Be careful about verb tense because the tense often shifts between past and present. A thorough re-reading is suggested.
L258 - This is the first mention of UAS-S4. The application of the wind tunnel results to the UAS is very unclear within the manuscript. Up to this point in the manuscript, the impression is that only wind tunnel tests were conducted for the morphing camber system applied to a single wing. The UAS aspect needs to be incorporated throughout, with a clear distinction between procedures and results from the wind tunnel, and procedures and results from the UAS simulation.
Answer 3.4
L177-L178: We replaced as suggested.
The Discussion section was divided in the revised paper into two sections, which are: “7. Discussion”, and “8. Application of results on the UAS-S4 Ehecatl”.
Comment 3.5
L277 - The theory that appears in this section should be moved earlier. Figures 22 to 24 seem to show results, so moving to the results section is more appropriate.
Answer 3.5
As mentioned also in the above answer, the Discussion section was divided in the revised paper into two other sections, which are: “7. Discussion”, and “8. Application of results on the UAS-S4 Ehecatl” for a better writing of the paper.
Comment 2.6
L300 - 304 - These equations should appear earlier in the methodology or theory sections.
Answer 2.6
These equations were not placed earlier in the paper because they referred to the fuselage, which was not explained earlier in the paper.
Round 2
Reviewer 2 Report
The responses are satisfactory.
I suggest some polishing of the language. some examples:
L40: 'consist on' should be 'consist of'
L233: 'It can thus be seen that the presence of the two morphing systems on the same wing does not generate any interferences with respect to their
separate presence on the wing.' --> sentence should be reworded.
L248: 'indicate' should be 'identify'
...
Also, all equations are blurry. Check fonts/format.
Author Response
Answers to reviewer 2 comments
We would like to thank very much to reviewer 2 for its comments. Please find below the comments of reviewer 2, and for each comment, our answers are given below in red colour - please see the attached pdf file.
Comment 1:
The responses are satisfactory.
I suggest some polishing of the language, some examples:
L40: 'consist on' should be 'consist of'
L233: 'It can thus be seen that the presence of the two morphing systems on the same wing does not generate any interferences with respect to their separate presence on the wing.' --> Sentence should be reworded.
L248: 'indicate' should be 'identify'
...
Answer 1:
We have done the following modifications suggested by reviewer 2:
L41: ‘consist on’ was replaced by ‘consist of’
L236: The sentence was changed into: ‘It can thus be seen that the presence of the MLE system does not generate any interferences on the MTE system and that no additional drag would occur due to the combination of the two systems (MLE and MTE) on the same wing.’
L252: ‘indicate’ was replaced by ‘identify’
Also, all equations are blurry. Check fonts/format.
The equations should not be blurry; we believe that the quality of equations was downgraded during submission process. The publisher of the paper will let us know if there will be problems for their editing in the journal format.
Many thanks once again to reviewer 2 for his comments that helped us to improve the paper.

Reviewer 3 Report
Dear Mr. Communier, Dr. Botez, and Dr. Wong:
Thank you for submitting the revised version of your manuscript. Unfortunately, there remain a large number of confusing sentences and grammar errors. I've highlighted all of them in the attached PDF.

Author Response
Answers to reviewer 3 comments
We would like to thank very much to reviewer 3 for its comments. Please find below the comments of reviewer 3, and for each comment, our answers are given below in red colour - please see the attached 'pdf' file.
Comment 1:
Dear Mr. Communier, Dr. Botez, and Dr. Wong:
Thank you for submitting the revised version of your manuscript. Unfortunately, there remain a large number of confusing sentences and grammar errors. I've highlighted all of them in the attached PDF.
Answer 1:
We corrected the sentences and the grammar errors according to yours comments in the attached pdf:
L14, L216: “however” was removed
L16: “This behavior” was changed to “The behavior of the aerodynamics performances from the MTE and the MLE”
L19: “reducing its drag” was replaced by “a drag reduction”
L23: “needed” was removed
L31, L65, L97, L128, L133, and L355: “can be” was replaced by “is”
L34, L41, L80, L100, L145, L198, and L248: “in order” was removed
L41: “on” was replaced by “of”
L48: “a” was added before “fish bone concept”
L61: “in order to increase also its lift on drag ratio” was replaced by “and lift to drag ratio”
L72: “divided each of them” was replaced by “each divided”
L73: the word “shape” was added
L81: “a lift also lower” was replaced by “a lower lift”
L84: “CL with CD” was replaced by “Cl with Cd”
L94: “which came” was removed
L118: The figure 3 was removed
L150: “Therefore” was removed
L152: “is resulting” was replaced by “results”
L153: “on” was replaced by “along”
L155, L156: “thus” was removed
L159: “but because of the fact” was removed
L201: “aerodynamic loading scales” was replaced by ”wing”
L217: “to be analyzed from Figure 16” was added
L234: The size (hight) of the figure was reduced to obtain the caption on the same page
L248: The sentence was rearranged to: “During the design of an aircraft, we seek to obtain a target lift value. To allow an analysis of the performance of the wing with the MCS, thus, it is preferred to represent the drag coefficient variation with the lift coefficient.”
L251, 263: Figure 20 was removed.
The sentence was rearranged into:”This idea allows to define its specific morphing configuration for drag minimization, for each flight case of an aircraft, characterized by a needed lift.”
L280: A description of the UAS-S4 was added:”The Unmanned Aerial System (UAS) S4 Ehécatl is an autonomous flight system. It includes an onboard camera, an autopilot and the sensors necessary for its operation. The UAS-S4 Ehécatl model was developed in Mexico by Hydra Technologies in 2002 and made its first flight in 2006. It is therefore a continuous development project for almost 15 years. The UAS-S4 Ehécatl is used by the Mexican army and police. As part of the research developed at the Research Laboratory in active control, avionics and aeroservoelasticity (LARCASE), we use it to develop new methods of analysis, design and manufacturing applied to morphing wings. The geometries for the UAS-S4 in this paper are in Table 5. ”
L288: Table 5 was added to include the geometrical parameters of the UAS-S4:
Table 5. Geometrical data of the UAS-S4
|
Wing Surface S |
2.9769 m² |
|
Root chord CR |
656.159 mm |
|
Wing span b |
4.19 m |
|
Wing Taper Ratio λ |
0.6057 |
L290: ”Equations (4) gives the aspect ratio (AR) of the wing and Equation (5) gives the value of the Oswald coefficient for a straight wing [1]. ” was removed
L294: “with” was replaced with “where the aspect ratio (AR) of the wing is”
“and the Oswald coefficient for a straight wing is” was added
L322: “which” was replaced by “This observation”
L324: “can therefore be” was replaced by “is”
L351: “can be” was replaced by “are”
L355: “on” was replaced by “in”
Many thanks once again to reviewer 3 for his c

Round 3
Reviewer 3 Report
Dear Mr. Communier, Dr. Botez, and Dr. Wong:
Thank you for making all the edits and modifications to your paper.